# Betulin Inhibits Lung Metastasis by Inducing Cell Cycle Arrest, Autophagy, and Apoptosis of Metastatic Colorectal Cancer Cells

**DOI:** 10.3390/nu12010066

**Published:** 2019-12-26

**Authors:** Yo-Han Han, Jeong-Geon Mun, Hee Dong Jeon, Ji-Ye Kee, Seung-Heon Hong

**Affiliations:** Department of Oriental Pharmacy, College of Pharmacy, Wonkwang-Oriental Medicines Research Institute, Wonkwang University, 460 Iksandae-ro, Iksan, Jeonbuk 54538, Korea; dygks1867@hanmail.net (Y.-H.H.); wjdrjs92@daum.net (J.-G.M.); alen0707@naver.com (H.D.J.)

**Keywords:** betulin, colorectal metastasis, apoptosis, autophagy, cell cycle arrest

## Abstract

Background: Colorectal cancer (CRC) is one of the diseases with high prevalence and mortality worldwide. In particular, metastatic CRC shows low probability of surgery and lacks proper treatment. In this study, we conducted experiments to investigate the inhibitory effect of betulin against metastatic CRC and related mechanisms. Methods: Water-soluble tetrazolium assay was used to determine the effect of betulin on metastatic CRC cell viability. Flow cytometry and TUNEL assay were performed to confirm whether betulin can induce apoptosis, autophagy, and cell cycle arrest. A lung metastasis mouse model was employed to estimate the anti-metastatic effect of betulin. Results: betulin decreased viability of metastatic CRC cells, including CT26, HCT116, and SW620 cell lines. Through PI3K/Akt/mTOR inactivation, betulin induced AMPK-mediated G0/G1 phase arrest and autophagy of CT26 and HCT116 cells. In addition, betulin occurred caspase-dependent apoptosis via the mitogen-activated protein kinase signaling pathway in metastatic CRC cells. Moreover, orally administered betulin significantly inhibited metastasis of CT26 cells to the lung. Conclusion: Our results demonstrate the anti-metastatic effect and therapeutic potential of betulin in metastatic CRC treatment.

## 1. Introduction

Colorectal cancer (CRC) is the second leading cause of death in the Western world in both males and females [1]. CRC incidence and mortality rates are increasing worldwide, especially in economically developed countries, which have the highest incidence rates [2]. Globally, 2.2 million new cases and 1.1 million deaths due to CRC are expected by 2030 [3]. Approximately 20% of CRC patients have metastasis at the time of diagnosis and treatment of metastatic CRC is considered challenging [4]. Given the poor survival prognosis of metastatic CRC, early phase detection of CRC and discovery of new therapeutic substances for treating metastatic CRC is important. 

Induction of cell cycle arrest and programmed cell death in cancer cells are efficient strategies in cancer treatment. There are two types of programmed cell death, apoptosis and autophagy. Apoptosis (type I programmed cell death) has several morphological and biochemical change characteristics, such as cell shrinkage, membrane blebbing, nuclear condensation, and DNA fragmentation [5]. During apoptosis, activation of caspases leads to cleavage of essential apoptotic proteins. Among the caspases, caspase-3 is the critical executioner of apoptosis, responsible for proteolytic cleavage of nuclear enzyme poly (ADP-ribose) polymerase (PARP) which plays an important role in DNA repair [6]. 

Autophagy (type II programmed cell death) also described as “self-eating,” is a lysosomal degradation response to degrading cellular proteins, organelles, and cytoplasm to sustain cellular metabolism under harsh conditions such as starvation and stress. When autophagy is triggered, mainly via AMP-activated protein kinase (AMPK) and PI3K/Akt/mTOR signaling pathways, the expression of LC3B protein—a double-membrane component of the autophagosome—is increased to form the autophagosome which fuses with the lysosome to remove damaged cell organelles [7]. Several cytotoxic drugs eliminate cancer cells by inducing excessive cellular damage to initiate autophagic cell death [8,9].

Nowadays several studies report the usefulness of natural compounds as anti-cancer agents [1]. Betulin, a naturally occurring pentacyclic triterpene (Figure 1A), is mainly found in the bark of birch species (Betulaceae). Betulin has several pharmacologic effects, including anti-inflammatory, anti-amnesic, and anti-osteoclastogenic effects [10,11,12]. Additionally, betulin shows a protective effect in hepatitis, kidney injury, and chronic obstructive pulmonary disease [13,14,15]. Although, betulin exhibits anti-cancer properties against diverse cancer cell lines, including leukemia, melanoma, gastric cancer, lung cancer, breast cancer, and prostate cancer, its effect on metastatic CRC cells has not been elucidated [16]. In this study, we investigated the anti-proliferative effect of betulin on metastatic CRC cells and related mechanisms.

## 2. Materials and Methods

### 2.1. Reagents

We purchased betulin from Chengdu Biopurify Phytochemicals Ltd. (Chengdu, China), cell counting kit (CCK)-8 from DoGen (Daejeon, Korea), compound C (CC) from MedChemExpress (Monmouth Junction, NJ, USA), and crystal violet solution from Sigma–Aldrich (St Louis, MO, USA).

### 2.2. Cell Culture

The murine CRC cell line colon 26 (CT26) and human CRC cell lines HCT116 and SW620 were purchased from Korean Cell Line Bank (Seoul, Republic of Korea). CT26 cells were maintained in Dulbecco’s modified Eagle’s medium. HCT116 and SW620 cells were cultured in RPMI 1640. The mediums contained 10% fetal bovine serum and 100 U/mL Penicillin-Streptomycin (Thermo Fisher Scientific, MA, USA).

### 2.3. Cell Viability Measurement

The viability of cells after betulin (0–8 μM) treatment was measured using the CCK-8 reagent. Cells were seeded in a 96-well plate (3 × 10^3^ cells/well/200 μL) and treated with betulin for 72 h. New medium containing CCK-8 was added to the plate, and the absorbance was measured using a microplate reader.

### 2.4. Colony Formation

Cells were seeded into a 12-well culture plate (5 × 10^2^ cells/well) and incubated with betulin for 7 days. The colonies were fixed with 3.7% formaldehyde for 30 min and washed using phosphate buffered saline (PBS). Colonies were stained using crystal violet solution (0.1%) for 20 min. The stained colonies were photographed after PBS washing.

### 2.5. Cell Cycle Distribution 

Cell cycle analysis was conducted using the Muse Cell Cycle Kit and Muse Cell Analyzer (MUSE, Millipore, Bedford, MA, USA). CT26 and HCT116 cells (5 × 10^5^ cells/well) in 6-well plates were treated with betulin (0–8 μM) for 24 h. Manufacturer protocols were followed for staining and analysis of propidium iodide (PI)-positive cells.

### 2.6. Real-Time Reverse Transcription Polymerase Chain Reaction (RT-PCR)

Total RNA was extracted using an RNA-spin^TM^ Total RNA Extraction Kit (iNtRon Biotech, Seoul, Republic of Korea) and reverse transcribed to cDNA using the High-Capacity cDNA Reverse Transcription Kit (Applied Biosystems, Foster City, CA, USA). Expression of target genes was quantified using the Power SYBR^®^ Green PCR Master Mix and Step-one Plus^TM^ Real-Time PCR Systems (Applied Biosystems, Foster City, CA, USA). The mouse primers for real-time RT-PCR were as follows: cyclin D1, 5′-TAGGCCCTCAGCCTCACTC-3′ (forward) and 5′-CCACCCCTGGGATAAAGCAC-3′ (reverse); cdk4, 5′-AGAGCTCTTAGCCGAGCGTA-3′ (forward) and 5′-TTCAGCCACGGGTTCATATC-3′ (reverse); and gapdh, 5′-GACATGCCGCCTGGAGAAAC-3′ (forward) and 5′-AGCCCAGGATGCCCTTTAGT-3′ (reverse). The human primers for real-time RT-PCR were as follows: Cyclin D1, 5′-ATGCCAACCTCCTCAACGAC-3′ (forward) and 5′-GGCTCTTTTTCACGGGCTCC-3′ (reverse); CDK4, 5′-GTGCAGTCGGTGGTACCTG-3′ (forward) and 5′-TTCGCTTGTGTGGGTTAAAA-3′ (reverse); GAPDH, 5′-TGCACCACCACCTGCTTAGC-3′ (forward) and 5′-GGCATGGACTGTGGTCATGAG-3′ (reverse).

### 2.7. Detection of Autophagy

Muse ^TM^ Autophagy LC3-antibody based kit (MUSE, Millipore, Bedford, MA, USA) was used to detect autophagy of cancer cells after incubation with betulin for 24 h. According to the manufacturer’s protocol, cells were permeabilized and incubated with the anti-LC3 Alexa Fluor 555-conjugated antibody for 30 min. Intracellular LC3 fluorescence was detected and analyzed using the Muse Cell Analyzer. 

### 2.8. Western Blot Analysis

PRO-PREP ^TM^ Protein Extraction Solution (iNtRon Biotech, Seoul, Korea) was used to extract total proteins from cells and tissues. Lysates were mixed with 5 × sample buffer volume and total proteins were separated using gel electrophoresis. Target proteins were detected with the following antibodies: Anti-phopho-AMPK, LC3-II, beclin-1, phospho-PI3K, phospho-Akt, phospho-mTOR, phospho-p38, phospho-ERK, phospho-JNK, AMPK, PI3K, PARP, caspase-3, caspase-9, Bcl-xL, and Bax (Cell Signaling, Danvers, MA, USA). Anti-Akt, p38, ERK, JNK, Bcl-2, GAPDH, cyclin D1, CDK4, and α-tubulin antibodies were purchased from Santa Cruz Biotechnology (CA, USA). Specific proteins were detected by secondary antibodies and visualized using the FluorChem M System (ProteinSimple, San Jose, CA, USA).

### 2.9. Measurement of Apoptosis

Apoptosis of betulin-treated cells was detected using terminal deoxynucleotidyl transferase dUTP nick end labeling (TUNEL) and annexin V assays as previously described [17]. TUNEL-positive cells were observed using a fluorescence microscope (Thermo Fisher Scientific, MA, USA), and annexin-positive cells were analyzed using the Muse^®^ Annexin V and Dead Cell Kit (Millipore, Billerica, MA, USA) in accordance with the recommended protocol.

### 2.10. In-Vivo Model of Lung Metastasis

Animal experimental methods were approved by Wonkwang University Institutional Animal Care and Use Committee (WKU18–25). BALB/c mice (5-week-old) were purchased from Samtaco Korea (Osan, Republic of Korea) and mice were housed in a laminar air-flow room. For in-vivo experiments, CT26 cells (2 × 10^5^ cells) were intravenously injected into the tail vein of mouse. Betulin (5 and 10 mg/kg) was orally administered to the mouse once a day until day 14. After sacrifice, the lungs were stained with Bouin’s solution and the number of colonies in the lungs was counted. To confirm the molecular mechanisms of anti-metastatic effect of betulin, lungs were harvested and stored at −80 °C.

### 2.11. Statistical Analysis

Statistical analyses (Student’s *t*-test and one-way ANOVA) were performed using the SPSS version 23.0 software (IBM Corporation, Chicago, IL, USA). Means and standard deviations of at least three experiments were calculated and *p* < 0.05 was considered statistically significant.

## 3. Results

### 3.1. Betulin Suppresses Viability of Metastatic CRC Cells

To investigate the viability of CRC cells after incubation with betulin, metastatic CRC cell lines CT26, HCT116, and SW620 were treated with betulin for 24–72 h. Betulin decreased viability of CT26, HCT116, and SW620 cells in a dose- and time-dependent manner (Figure 1B–D). In addition, betulin inhibited the colony formation of CT26 and HCT116 cells (Figure 1E). When compared with the control group, betulin significantly decreased percentages of colony formation in CT26 and HCT116 cells to 68.1% and 88% (2 μM), 55.11% and 82.49% (4 μM), and 40.21% and 78.64% (8 μM), respectively (Figure 1F).

### 3.2. Betulin Promotes Cell Cycle Arrest in G0/G1 Phase of CRC Cells Through AMPK Activation

To investigate whether the anti-proliferative effect of betulin is due to cell cycle arrest, flow cytometry was performed to determine cell cycle phase distributions of betulin-treated CRC cells. Betulin induced G0/G1 phase arrest by increasing the percentage of CT26 cells from 53% to 60% (Figure 2A,B). Similarly, the percentage of HCT116 cells in the G0/G1 phase increased from 54% to 63% (Figure 2C,D). Reduced expression of cyclin D1/CDK4 complex is one of the causes of G0/G1 phase arrest [18]. Real-time RT-PCR results showed that betulin treatment significantly decreased mRNA expressions of cyclin D1 and CDK4 in both CT26 and HCT116 cells (Figure 2E,F). Additionally, betulin increased AMPK phosphorylation in CT26 and HCT116 cells in a dose-dependent manner (Figure 2G,H).

Recently, studies showed that AMPK activation induced G0/G1 phase arrest by decreasing expression of cyclin D1 and CDK4 in diverse cancer cells [19,20]. To confirm whether betulin-induced cell cycle arrest is mediated by AMPK activation, we treated AMPK inhibitor CC with the highest concentration of betulin to CRC cells. Treatment with CC (10 μM) for 24 h did not change viability of CT26 and HCT116 cell lines (Figure 3A,B). Moreover, CC treatment suppressed G0/G1 phase arrest of both CRC cells after betulin treatment (Figure 3C). Additionally, CC treatment blocked AMPK activation and betulin-induced cell cycle arrest in both cell lines (Figure 3D,E). In the mRNA and protein expression levels, betulin-induced down-regulation of cyclin D1 and CDK4 was recovered by CC treatment in CT26 cells (Figure 3F,G). These results suggest that betulin induces G0/G1 phase arrest of metastatic CRC cells via cyclin D1 and CDK4 down-regulation through AMPK activation.

### 3.3. Betulin Increases Autophagy of CRC Cells Via PI3K/Akt/mTOR and AMPK Signaling Pathway

AMPK activation is an important factor associated with autophagy in cancer cells [21]. Autophagy is mainly mediated via the PI3K/Akt/mTOR and AMPK/mTOR signaling pathways [22]. Since we observed AMPK activation and changes in cell viability by betulin, we investigated whether betulin can cause autophagy. We confirmed that betulin induces autophagy of both CRC cells by LC3 detection (Figure 4A). The autophagy ratio was doubled by the highest concentration of betulin treatment. (Figure 4B,C). Western blot analysis showed that betulin increased the expression of LC3-II and beclin in both CRC cell lines by reducing phosphorylation of PI3K, Akt, and mTOR (Figure 4D,E). To determine whether betulin-induced AMPK activation was also related to autophagy induction, CC (AMPK inhibitor) was used. Blockage of AMPK activation by CC treatment decreased LC3 intensity which indicates autophagic cell death in the betulin-treated CRC cells (Figure 5A–C). Increased expression of LC3-II and beclin-1 were also reduced by AMPK inactivation (Figure 5D,E). These results indicate that betulin can promote autophagic cell death of CRC cells and the betulin-induced autophagy was mediated via PI3K/AKT/mTOR and AMPK signaling pathways.

### 3.4. Betulin Induces Apoptosis of CRC Cells by Decreasing Phosphorylation of MAPKs 

Although betulin can induce cell cycle arrest and autophagy of metastatic CRC cells, it did not reach the degree of cell viability reduction as was seen in the cell viability assay results. Thus, we carried out TUNEL and annexin V assays to investigate whether betulin can induce apoptosis of CRC cells. As shown in Figure 6A,B, TUNEL and annexin V-positive cells were increased by betulin treatment, especially 8 μM of betulin, which significantly increased percentages of apoptotic cells to 17.22% ± 1.82% and 13.79% ± 2.16% in CT26 and HCT116 cells, respectively (Figure 6C,D). Apoptosis-related proteins, including expression of Bax and cleavage of caspase-3, caspase-9, and PARP were increased by betulin, whereas expressions of Bcl-2 and Bcl-xL were decreased in both CRC cell lines (Figure 6E,F). To identify molecular mechanisms involved in betulin-induced apoptosis, phosphorylation of MAPKs (ERK, JNK, and p38) was determined by western blot analysis. The MAPK signaling pathway is related to cancer cell survival and several MAPK inhibitors have potential therapeutic utility in cancer by inducing apoptosis [23,24]. In CT26 and HCT116 cells, betulin suppressed phosphorylation of ERK, JNK, and p38 in a dose-dependent manner (Figure 6G,H). Therefore, betulin can induce ERK, JNK, and p38-mediated apoptosis of metastatic CRC cells.

### 3.5. Betulin Inhibits Lung Colonization of CT26 Cells in Mice

To investigate the effect of betulin on lung metastasis in the mouse model, CT26 cells were inoculated into the tail vein and 5 and 10 mg/kg of betulin were orally administered to the mouse. Two weeks after the treatment, the lung metastasis of CT26 cells was reduced compared with the control group (Figure 7A). The number of nodules were 184.67 ± 8.75 and 136.14 ± 11.12 in the 5 and 10 mg/kg betulin-administered mice, respectively, and 223.27 ± 15.76 in the control group (Figure 7B). To determine whether cell cycle arrest, autophagy, and apoptosis contributed to the inhibitory effect of betulin on lung metastasis, as observed in the in-vitro experiments, western blot analysis was performed to detect the expression of related proteins. In the betulin-treated group, protein expression levels of cyclin D1 and CDK4 were decreased according to AMPK activation (Figure 7C). As expected, betulin-induced autophagic cell death of CT26 cells by increasing LC3-II and beclin expressions through PI3K/AKT/mTOR inactivation in the lung tissues (Figure 7D). Betulin administration caused an increase in cleavage of caspase-3, caspase-9, and PARP and regulated the expression of Bcl-2, Bcl-xL, and Bax to cause apoptosis of cells (Figure 7E). Similar to the in-vitro experiments, phosphorylation of ERK, JNK, and p38 was also reduced in the betulin treatment group (Figure 7F). These results indicate that betulin can suppress the lung metastasis of CRC cells by inducing cell cycle arrest, apoptosis, and autophagy.

## 4. Discussion

Betulin is contained in many plants such as *Zizyphus mauritiana*, *Ziziphus vulgaris* var. *spinosus*, *Nelumbo nucifera*, and *Viscum album* var. *coloratum*. Especially, betulin and its derivatives selectively shows an inhibitory effect on the cancer cells with non-toxic concentration towards normal cells. Lup-20(29)-en-3β,28-di-yl-nitrooxy acetate, a derivative of betulin, induces G0/G1 phase arrest, apoptosis, and autophagy in MCF-7 breast cancer cells [25]. 3,28-di-(2-nitroxy-acetyl)-oxy-betulin also induces cell cycle arrest and apoptosis in hepatocarcinoma Huh7 cells [26]. Our results also showed that betulin significantly decreased viability of metastatic CRC cells in a dose- and time-dependent manner and induced G0/G1 phase arrest, apoptosis, and autophagy of CT26 and HCT116 cells.

AMPK is one of the key factors to suppress cancer progression because it can regulate cell cycle and autophagy [27]. Activation of AMPK can induce G0/G1 phase cell cycle arrest by downregulating cyclin D1 in CRC and myeloma cells [28,29]. Moreover, the AMPK activator hernandezine induces autophagic cell death in various human cancer cells such as HeLa, A549, MCF-7, PC3, HepG2, Hep3B, and H1299 [30]. In this study, betulin (8 μM) induced G0/G1 phase arrest and autophagy of CRC cells. As expected, betulin treatment increased AMPK phosphorylation and constantly down-regulated expression of cell cycle-related factors and autophagic markers. AMPK inactivation by CC in betulin-treated CRC cells decreased cell cycle arrest and autophagy. These results prove that betulin-induced G0/G1 arrest and autophagy are mediated via AMPK activation.

Although betulin induces cell cycle arrest and autophagy, it has a meagre effect compared to reduction of CRC cell viability. Several studies have reported that betulin shows selective inhibitory effects on proliferation of several cancer cells by inducing caspase-3- and caspase-9-mediated apoptosis [31,32,33]. In addition, betulin can regulate expressions of Bcl-2 and Bax in gastric and cervical cancer cells [33,34]. TUNEL and annexin V assay results demonstrated that betulin can induce apoptosis of CT26 and HCT116 cells.

MAPKs family members such as ERK, JNK, and p38 regulate diverse biological processes. They are involved in cell survival, apoptosis, differentiation, and gene expression through phosphorylation of their substrates upon stimulation [24]. MAPKs can positively or negatively regulate apoptosis against extracellular stress [35]. Several studies reported that inhibition of MAPKs phosphorylation decreases proliferation of CRC cells. A p38 inhibitor, FR167653 induced caspase-dependent apoptosis in human CRC cell lines DLD-1 and SW480 [36]. FR180204, an ERK inhibitor, combined with Akt inhibitor reduces viability of CRC cells and increases apoptosis of CRC cells. In addition, JNK inhibitor AS601245 decreases proliferation of Caco-2 cells [24]. A study reported that betulin decreased phosphorylation of ERK, JNK, and p38 in RAW 264.7 cells [10]. Hence, the effect of betulin on the phosphorylation of MAPKs was confirmed and betulin treatment suppressed ERK, JNK, and p38 phosphorylation in the metastatic CRC cells. These results suggest that the apoptotic effect of betulin on CRC cells might be mediated via MAPKs signaling pathway.

To investigate the anti-metastatic effect of betulin in our in-vivo experiments in mouse models, the maximum concentration of betulin for oral administration was set at 10 mg/kg. Pharmacokinetic research in rats has showed that daily intraperitoneal administration of 60 mg/kg of betulin for 28 days does not result in toxicity [37]. Oral administration of 5 and 10 mg/kg betulin improved the scopolamine-induced amnesia in mouse models, and intraperitoneal administration of betulin (10 and 20 mg/kg) improved autoimmune hepatitis in mice [11,38]. Additionally, betulin administration (10 mg/kg) significantly decreased lung metastasis of CT26 cells. Similarly, in our in-vitro experiment, betulin regulated factors related to cell cycle and autophagy by increasing AMPK phosphorylation and suppressing the PI3K/AKT/mTOR signaling pathway. In addition, apoptotic proteins and activation of MAPKs were regulated in the betulin-treated group.

## 5. Conclusions

This study demonstrated that betulin exhibits inhibitory effects on colorectal metastasis by inducing cell cycle arrest and autophagy in metastatic CRC via AMPK and PI3K/Akt/mTOR signaling pathways. Moreover, betulin induces apoptosis through MAPKs inactivation. Based on these results, betulin may contribute in the treatment of metastatic CRC as a novel therapeutic agent.

## Figures and Tables

**Figure 1 nutrients-12-00066-f001:**
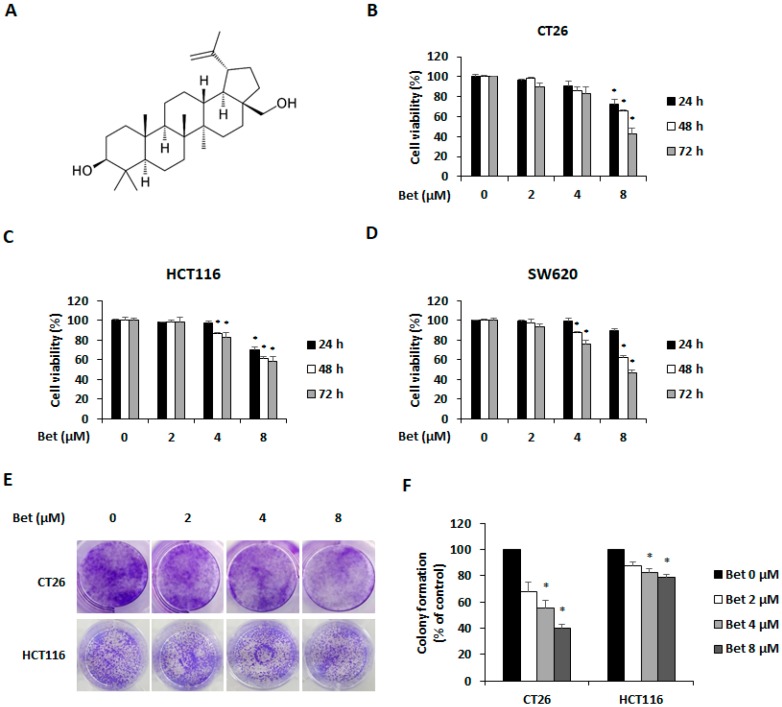
Betulin inhibits viability and colony formation of metastatic colorectal cancer cells. Chemical structure of betulin (**A**). Viability of betulin-treated CT26 (**B**), HCT116 (**C**), and SW620 (**D**) cells. (**E**) Colony formation of CT26 and HCT116 cells with betulin treatment for 7 days. (**F**) Counting of colony formation. Images were captured using phase contrast microscope (magnification, 200×). Data are means ± standard deviation of three independent experiments. * *p* < 0.05.

**Figure 2 nutrients-12-00066-f002:**
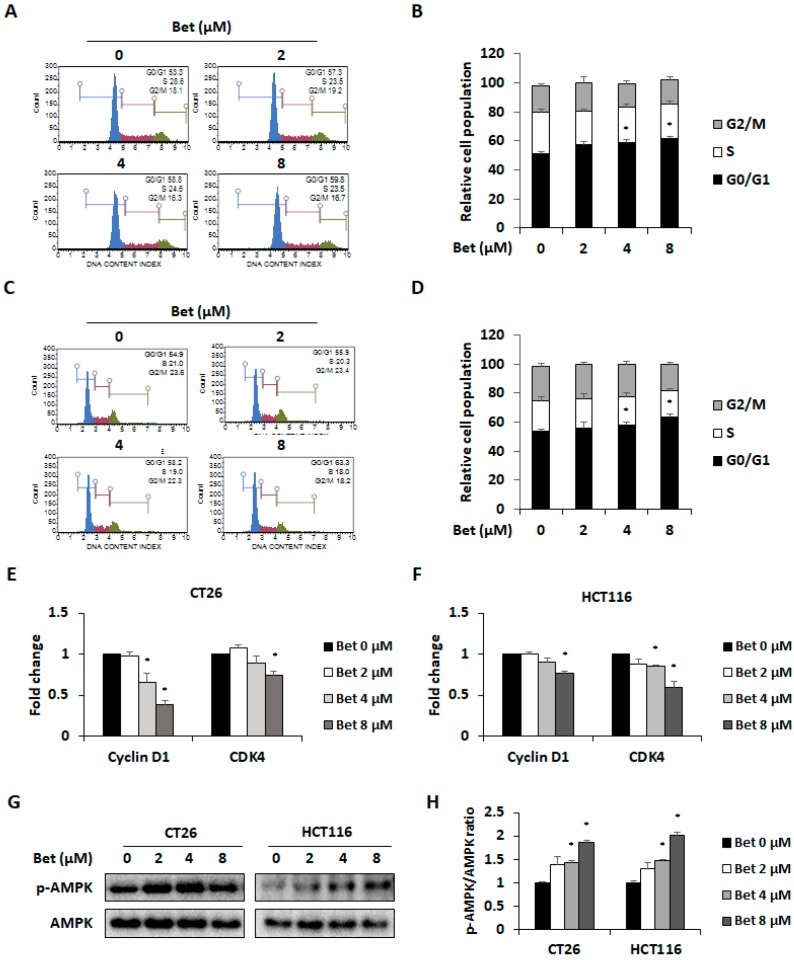
Betulin induces G0/G1 arrest through cyclin D1 and CDK4 downregulation in metastatic colorectal cancer cells. Cell cycle phase distribution (**A**) and relative cell population (**B**) of betulin-treated CT26 cells as determined by flow cytometry. Cell cycle phase distribution (**C**) and relative cell population (**D**) of betulin-treated HCT116 cells. Expression of cyclin D1 and CDK4 in betulin-treated CT26 (**E**) and HCT116 (**F**) cells. AMPK phosphorylation detected in betulin-treated CT26 (**G**) and HCT116 (**H**) cells. Images represent three independent experiments. Data are means ± standard deviation of three independent experiments. * *p* < 0.05.

**Figure 3 nutrients-12-00066-f003:**
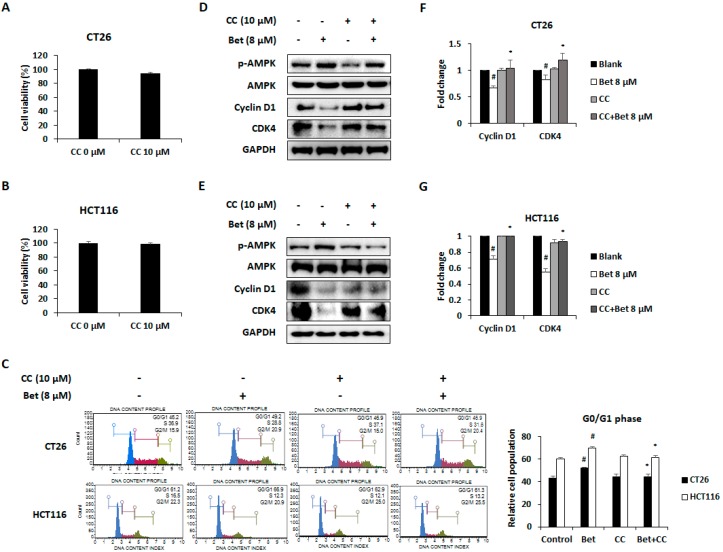
G0/G1 phase arrest by betulin in metastatic colorectal cancer cells is mediated via AMPK activation. Cell viability of CT26 (**A**) and HCT116 (**B**) cells after compound C (CC) treatment for 24 h. (**C**) Changes of cell cycle phase in betulin-treated CT26 and HCT116 cells by CC treatment. Images represent three independent experiments. AMPK phosphorylation and protein expression of cyclin D1 and CDK4 were determined in betulin and CC-treated CT26 (**D**) and HCT116 (**E**) cells. mRNA expression levels of cyclin D1 and CDK4 in CT26 (**F**) and HCT116 (**G**) cells with betulin and CC treatment for 24 h. Data are means ± standard deviation of three independent experiments. # *p* < 0.05 versus control and * *p* < 0.05 versus betulin alone.

**Figure 4 nutrients-12-00066-f004:**
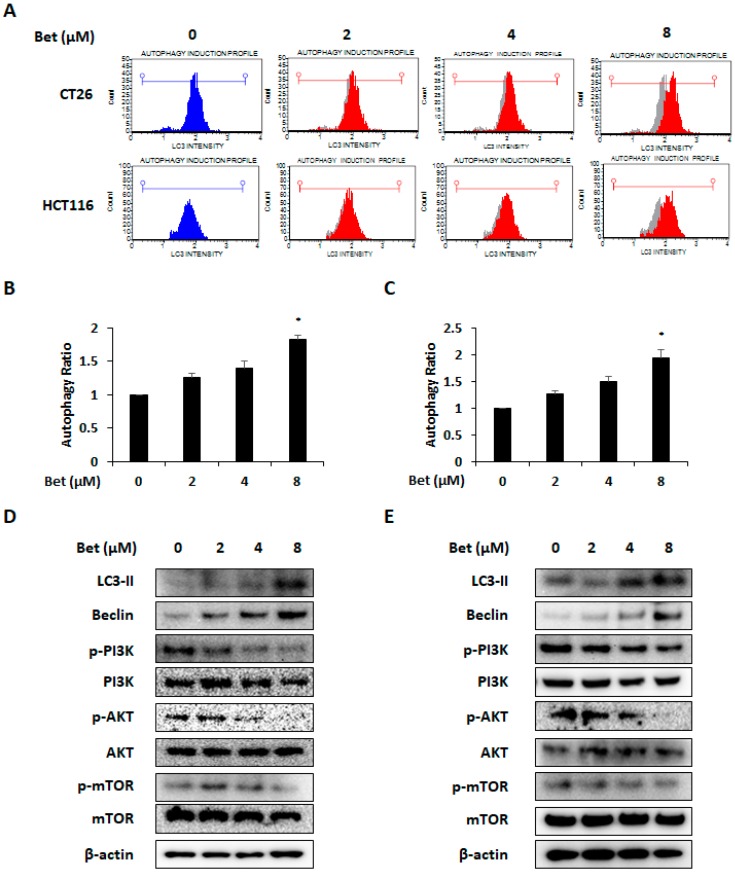
Betulin promotes autophagy of metastatic colorectal cancer cells via PI3K/Akt/mTOR signaling pathway. (**A**) LC3-II expression in betulin-treated CT26 and HCT116 cells was analyzed by flow cytometry. Autophagy ratio of CT26 (**B**) and HCT116 (**C**) cells with betulin treatment for 24 h. Protein bands representing LC3-II, beclin-1, p-PI3K, p-Akt, and p-mTOR in CT26 (**D**) and HCT116 (**E**) cells as detected by western blot analysis. Images represent three independent experiments. Data are means ± standard deviation of three independent experiments. * *p* < 0.05.

**Figure 5 nutrients-12-00066-f005:**
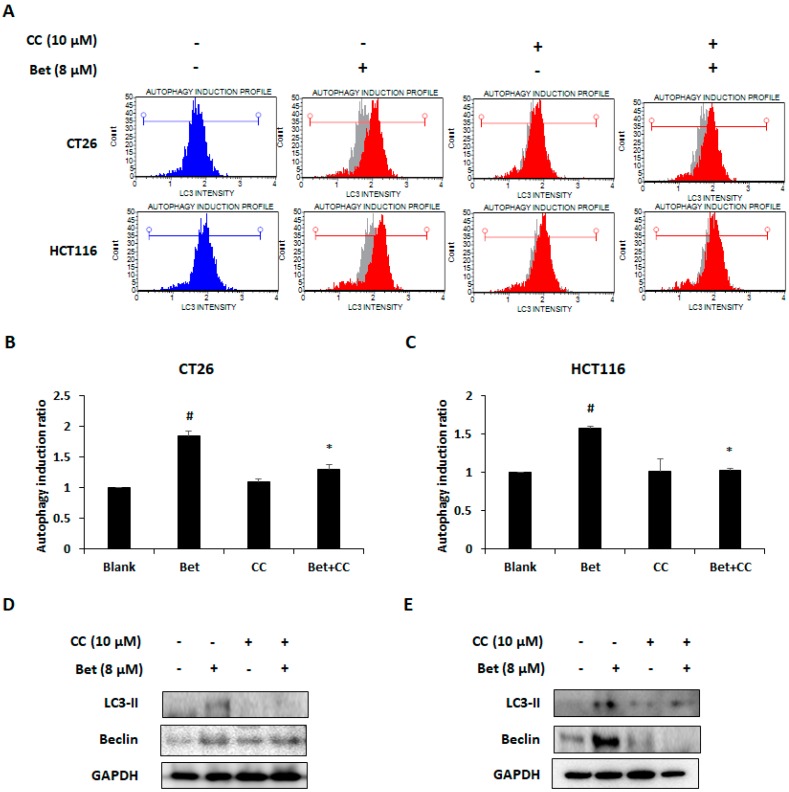
Betulin-induced AMPK activation is related to autophagy of metastatic colorectal cancer cells. (**A**) LC3 detection in betulin-treated CT26 and HCT116 cells after compound C (CC) treatment as detected by flow cytometry. Autophagy induction ratio of CT26 (**B**) and HCT116 (**C**) cells after betulin and CC treatment. Autophagy induction ratio was calculated by Muse Cell Analyzer. Expression changes of LC3-II and beclin-1 in betulin-treated CT26 (**D**) and HCT116 (**E**) cells. Images represent three independent experiments. Data are means ± standard deviation of three independent experiments. # *p* < 0.05 versus blank and * *p* < 0.05 versus betulin alone.

**Figure 6 nutrients-12-00066-f006:**
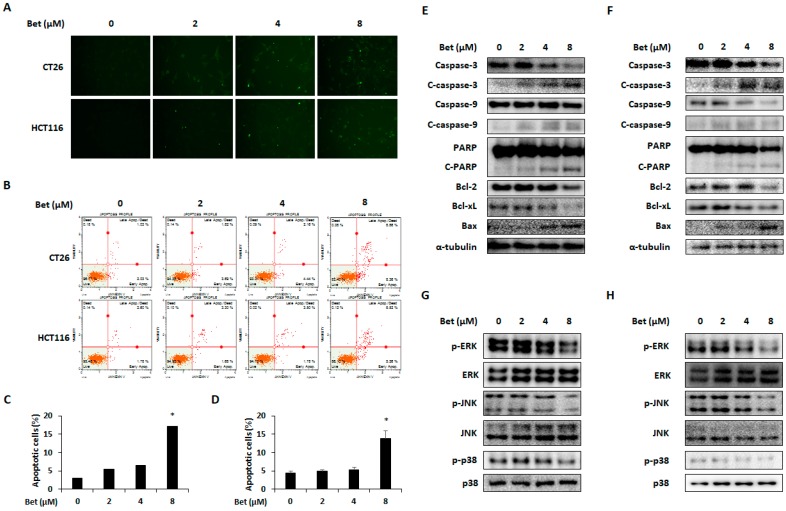
Betulin induces caspase-dependent apoptosis of metastatic colorectal cancer cells via MAPK signaling pathways. Apoptosis of CT26 and HCT116 cells by betulin as detected by TUNEL assay (**A**) and Annexin V assay (**B**). Percentages of apoptotic cells in betulin-treated CT26 (**C**) and HCT116 (**D**) cells. Apoptotic proteins in betulin-treated CT26 (**E**) and HCT116 (**F**) cells. Phosphorylation of ERK, JNK, and p38 in CT26 (**G**) and HCT116 (**H**) cells with betulin treatment. Images represent three independent experiments. Data are means ± standard deviation of three independent experiments. * *p* < 0.05.

**Figure 7 nutrients-12-00066-f007:**
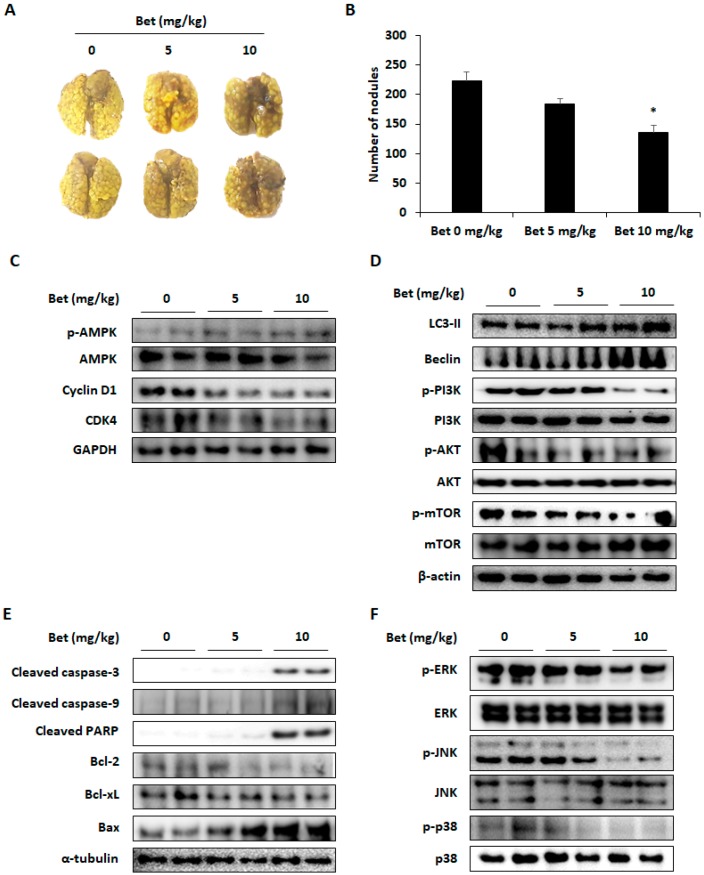
Betulin decreases lung metastasis of CT26 cells through cell cycle arrest, autophagy, and apoptosis by regulating AMPK, PI3K/Akt/mTOR, and MAPK signaling pathways. (**A**) Lung metastasis of CT26 cells was photographed at 14 days after inoculation. Images represent three independent experiments. (**B**) Number of nodules. (**C**) AMPK phosphorylation and expression of cyclin D1 and CDK4. (**D**) The level of autophagy-associated proteins. (**E**) Expression of apoptosis-related proteins in lung tissues as measured by western blot analysis. (**F**) The expression levels of p-ERK, p-JNK, and p-p38. Data are means ± standard deviation of three independent experiments. * *p* < 0.05.

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
