# Peer review of "Betulin Inhibits Lung Metastasis by Inducing Cell Cycle Arrest, Autophagy, and Apoptosis of Metastatic Colorectal Cancer Cells"

_nutrients, 2019, doi:10.3390/nu12010066_

Round 1

Reviewer 1 Report

Yo-Han Han et al. describe in the manuscript “Betulin inhibits colorectal metastasis by inducing cell cycle arrest, autophagy and apoptosis of metastatic colorectal cancer cells” the effect and mechanisms of betulin action on metastatic colorectal cancer cells. The study addresses a very focused specific question and the authors performed a solid work resolving this question using well established basic methods of cellular and molecular biology. In addition, they validated in vitro mechanistic findings in vivo by performing lung colonization assays with a metastatic colorectal cancer cell line. Although the results are clear, the conclusions including the title should be tuned down, as the type of in vivo assay addresses only late aspects of metastasis. In accordance with this, their findings rather indicate an influence of betulin in the outgrowth of metastatic nodules, but do not address early essential steps of metastatic spreading, such as invasive capacity.

Based on this, some minor revisions that includes a more moderate statement on the metastasis-related findings of the study.

Specific comments listed:

Titel: as mentioned above tune down to “late aspects of metastasis” Titel: it is incorrect to say colorectal metastasis, per definition it marks “colorectal” as the site of metastasis. Correctly it is lung metastasis of colorectal cancer. Figure 2 and 3: The figures would gain on structuring if Fig.3A would be moved to Fig2, and Fig 3 would be dedicated only for the effect of AMPK inhibitor. Fig 3: The CC study misses the effect of CC on cell viability and colony formation capability. Even if this is very moderate, it would also support the later reasoning to look at apoptosis in addition to G0/G1 arrest and autophagy. Fig 3: Quantification of cell cycle phases would be highly recommended, similarly as it is shown in Fig 2 B and D. Fig:3 Panels D and E are mixed up with panels F and G. Fig 5: B and C significance marking is missing. In addition, there is no description (neigther results, nor Mat&Meth) how autophagy induction ratio was calculated. Chapter 3.5 title: formulation is incorrect (see above). Suggested title is “Betulin inhibits lung colonization of CT26 cells in mice”. 7: Panel E and F are mixed up. There are several disturbing minor errors of English grammar. Furthermore, the word “occurs” is repeatedly used incorrectly instead of “causes”. Overall, English editing is highly recommended for better understanding.

Author Response

Reviewer 1

Title: as mentioned above tune down to “late aspects of metastasis” Title: it is incorrect to say colorectal metastasis, per definition it marks “colorectal” as the site of metastasis. Correctly it is lung metastasis of colorectal cancer.

→ Thanks for your kindly comment. Title was changed to ‘Betulin inhibits lung metastasis by inducing cell cycle arrest, autophagy, and apoptosis of metastatic colorectal cancer cells.’

Figure 2 and 3: The figures would gain on structuring if Fig.3A would be moved to Fig2, and Fig 3 would be dedicated only for the effect of AMPK inhibitor. Fig 3: The CC study misses the effect of CC on cell viability and colony formation capability. Even if this is very moderate, it would also support the later reasoning to look at apoptosis in addition to G0/G1 arrest and autophagy. Fig 3: Quantification of cell cycle phases would be highly recommended, similarly as it is shown in Fig 2 B and D.

→ As your comment, Fig. 3A and 3B were moved to Fig. 2G and 2H. Also, cell viability of CT26 and HCT116 cells after compound C (CC) treatment for 24 h was presented to Fig. 3A and 3B. Quantification of cell cycle phases was added to Fig. 3C.

Fig. 3: Panels D and E are mixed up with panels F and G.

→ As your comment, it has been corrected.

Fig 5: B and C significance marking is missing. In addition, there is no description (neigther results, nor Mat&Meth) how autophagy induction ratio was calculated.

→ Significance marking was added to Fig. 4B and C. Autophagy induction ratio in CRC cells after botulin treatment was calculated by Muse analyzer based on detection of LC3-II expression.

Chapter 3.5 title: formulation is incorrect (see above). Suggested title is “Betulin inhibits lung colonization of CT26 cells in mice”.

→ As your comment, it has been corrected.

7: Panel E and F are mixed up. There are several disturbing minor errors of English grammar. Furthermore, the word “occurs” is repeatedly used incorrectly instead of “causes”.

→ As your comment, it has been corrected.

Overall, English editing is highly recommended for better understanding.

→ Language editing was completed by Editage. Please see the certificate.

Reviewer 2 Report

Dear authors,

I read with interest your manuscript, but I think you should improve it.

Major points:

1) The authors have to improve the introduction as follow:

- lines 29-31: "Colorectal cancer (CRC) is the third most diagnosed malignancy and is one of the main causes of cancer-related deaths in the world. Incidence and mortality rates of CRC are increasing in worldwide and especially become more serious in developing countries." The authors have to delete this paragraph and add the following one.

"The colorectal cancer (CRC) represents the second leading cause of death in the Western world in both males and females (Malfa et al., 2019).  Incidence and mortality rates of CRC are increasing in worldwide and especially the highest incidence rates are found in economically developed countries (Acquaviva et al., 2016)."

-Please add the suggested references:

Malfa et al., 2019, Int. J. Mol. Sci. 2019, 20, 2723;  doi:10.3390/ijms20112723

Acquaviva et al., 2016  doi.org/10.3892/or.2016.5035

-Line 52 Add this sentence and the suggested reference.

"Nowaday several literature data report the usefulness of natural compounds as anti-cancer agents (Malfa et al., 2019). Among these, betulin...line 53.

-Lines 69-71: the authors should better describe the composition of culture medium, adding %FBS and antibiotics. In addition, they must report the betulin concentrations used for the treatment.

-Line 75: add λ 

-Please delete figure A and improve the quality of all figures. 

Minor points:

line 19 Lung metastasis mouse model was employed to estimate the anti-metastatic effect of betulin in vivo.  line 57 exhibits lines 78-79 CT26 and HCT116 cells (5 × 105 cells/well) in 6-well plates were treated with betulin (add concentrations) for 24 h.

Author Response

Reviewer 2

Major points:

1) The authors have to improve the introduction as follow:

- lines 29-31: "Colorectal cancer (CRC) is the third most diagnosed malignancy and is one of the main causes of cancer-related deaths in the world. Incidence and mortality rates of CRC are increasing in worldwide and especially become more serious in developing countries." The authors have to delete this paragraph and add the following one.

"The colorectal cancer (CRC) represents the second leading cause of death in the Western world in both males and females (Malfa et al., 2019).  Incidence and mortality rates of CRC are increasing in worldwide and especially the highest incidence rates are found in economically developed countries (Acquaviva et al., 2016)."

-Please add the suggested references:

Malfa et al., 2019, Int. J. Mol. Sci. 2019, 20, 2723;  doi:10.3390/ijms20112723

Acquaviva et al., 2016  doi.org/10.3892/or.2016.5035

-Line 52 Add this sentence and the suggested reference.

"Nowaday several literature data report the usefulness of natural compounds as anti-cancer agents (Malfa et al., 2019). Among these, betulin...line 53.

→ As your comment, it has been described.

-Lines 69-71: the authors should better describe the composition of culture medium, adding %FBS and antibiotics. In addition, they must report the betulin concentrations used for the treatment.

→ The mediums contained 10% fetal bovine serum and 100 U/mL Penicillin-Streptomycin. It has been described in Line 70-72.

-Line 75: add λ

→ As your comment, it has been added.

-Please delete figure A and improve the quality of all figures.

→ As your comment, it has been edited.

Minor points:

line 19 Lung metastasis mouse model was employed to estimate the anti-metastatic effect of betulin in vivo. 

line 57 exhibits lines 78-79 CT26 and HCT116 cells (5 × 105 cells/well) in 6-well plates were treated with betulin (add concentrations) for 24 h.

→ Thanks for your kindly comment. It has been corrected.

Round 2

Reviewer 2 Report

The authors improved the manuscript as suggested